Testosterone and estradiol affect adolescent reinforcement learning

Kohne Sina sina.korf@uni-hamburg.de
Diekhof Esther K.
Faculty of Mathematics, Informatics and Natural Sciences, Department of Biology, Institute of Animal Cell and Systems Biology, Neuroendocrinology and Human Biology Unit, Universität Hamburg , Hamburg , Germany
Okpala Charles
Electronic publication date: 2022 Feb 3
Publication date: 2022
Volume: 10
Electronic Location ID: e12653
Received 2021 Apr 28; Accepted 2021 Nov 29
Copyright: ©2022 Kohne and Diekhof
Copyright year: 2022
Copyright holder: Kohne and Diekhof
License: This is an open access article distributed under the terms of the Creative Commons Attribution License, which permits unrestricted use, distribution, reproduction and adaptation in any medium and for any purpose provided that it is properly attributed. For attribution, the original author(s), title, publication source (PeerJ) and either DOI or URL of the article must be cited.
License URL: https://creativecommons.org/licenses/by/4.0/

Keywords: Adolescence, Learning, Reward, Estradiol, Testosterone

Funding: The authors received no funding for this work.

==============================
During adolescence, gonadal hormones influence brain maturation and behavior. The impact of 17β-estradiol and testosterone on reinforcement learning was previously investigated in adults, but studies with adolescents are rare. We tested 89 German male and female adolescents (mean age ± sd = 14.7 ± 1.9 years) to determine the extent 17β-estradiol and testosterone influenced reinforcement learning capacity in a response time adjustment task. Our data showed, that 17β-estradiol correlated with an enhanced ability to speed up responses for reward in both sexes, while the ability to wait for higher reward correlated with testosterone primary in males. This suggests that individual differences in reinforcement learning may be associated with variations in these hormones during adolescence, which may shift the balance between a more reward- and an avoidance-oriented learning style.

Introduction

Sex hormones have a great impact on adolescent (neuro-) physiological maturation. With the onset of puberty at 9 to 10 years in girls and 10 to 12 years in boys, respectively, sex hormone level increases rapidly (Peper & Dahl, 2013). Sex hormone levels are regulated via the reproductive hypothalamic-pituitary-gonadal axis initiated by the secretion of hypothalamic gonadotropin releasing hormone (GnRH). GnRH thereby stimulates the synthesis and secretion of luteinizing hormones and follicle stimulating hormones in the pituitary, which in turn contribute to the maturation of the gonads and sex hormone secretion (Sisk & Foster, 2004).

The rising sex hormone level during adolescence significantly contributes to pubertal development. With attainment of sexual maturity, sex hormones maintain reproductive function (Sisk & Foster, 2004). Neurophysiological investigations demonstrated a different impact of testosterone and 17β-estradiol (E2) on brain maturation. Testosterone is related to an increase of global white and gray matter volume in male adolescents (Peper et al., 2009; Peper et al., 2011), whereas in female adolescents E2 may be negatively associated with gray matter volume (Peper et al., 2009). Further, E2 seems to predict white matter growth across the entire brain in both sexes (Herting et al., 2014). Moreover, neurophysiological developmental changes during adolescence could be better explained by hormonal and pubertal development (measured by the Pubertal Development Scale or Tanner Stages) than by chronological age (Herting et al., 2014; Wierenga et al., 2018).

Sex hormones are very important when it comes to behavior and cognitive function in animals and humans. Besides the impact of E2 and testosterone on adolescent reward-related risk-taking (i.a. Op De Macks et al., 2016), an influence on reward-related learning and cognition has been assumed as well (Diekhof, 2018; Hamson, Roes & Galea, 2016). In adult women, E2 may promote verbal memory and fluency (Hamson, Roes & Galea, 2016). In gonadectomized male and female rats, E2 was found to improve learning and memory even after physiological or psychological stressors (Hamson, Roes & Galea, 2016; Khaleghi et al., 2021). Moreover, studies with castrated male rats suggested that learning may be improved by testosterone treatment (Spritzer et al., 2011). In healthy older men, a short-term testosterone administration improved cognitive performance significantly (Cherrier et al., 2001). Findings from children (6 to 9 years) further showed a relationship between moderate testosterone levels and an average intelligence (IQ between 70 and 130), whereas enhanced testosterone concentrations were related to high (IQ > 130), but also low intelligence (IQ < 70) (Ostatníková et al., 2007). Other studies also reported enhanced testosterone concentrations in children and young adolescents (6 to 13 years) with learning disabilities compared to peers without impairments (Kirkpatrick et al., 1993). Given this evidence, one may assume that during early adolescence balanced testosterone concentrations may be important for efficient cognitive processing.

One way for sex hormones to modulate aspects of reward processing and reinforcement learning is through the neurotransmitter dopamine. Both estradiol and testosterone can act as natural dopamine-agonists, which promote dopamine release and dopaminergic transmission through various physiological mechanisms (Becker, 1990; Castner, Xiao & Becker, 1993; Pasqualini et al., 1995; Sinclair et al., 2014). This is in so far important, since dopamine plays a crucial role in reinforcement learning and determines how proficient individuals learn from positive or negative action outcomes. It has been assumed that changes in dopamine following so called reward prediction errors possibly act via two anatomically distinct pathways in the mesocorticolimbic dopamine system (Maia & Frank, 2011). The activation of the Go pathway after the dopamine burst that follows unexpected reward entails in a repetition of the same action. In turn, activation of the NoGo pathway results from a dip in the tonic dopamine level, which facilitates learning from unexpected reward reduction, omission, or even punishment. This optimally promotes an adaption of action choice to maximize overall reward (Frank, Seeberger & O’Reilly, 2004).

A study using a response time (RT) adaption task, the so-called “clock task”, demonstrated this relation between dopamine and reinforcement learning by showing that patients with Parkinson’s disease, but pharmacologically normalized dopamine concentration, were better in the Go learning aspect of the task. These medicated patients thereby showed an enhanced ability to speed up for a reward (i.e., better ability to acquire a higher reward through quickly responding after trial onset). In comparison, in an unmedicated state and thus with pathologically lowered dopamine, the same patients, demonstrated a better NoGo learning ability. This was indicated by an increased capacity to slow down responding for reward maximization (i.e., enhanced capacity to wait for higher reward) (Moustafa et al., 2008).

With the same task, Diekhof and colleagues characterized the impact of periodically fluctuating sex hormones in women on Go as opposed to NoGo learning ability. They compared the RT adaption during three different menstrual cycle phases of late luteal phase, luteal phase and early follicular phase. During the late follicular phase E2 is high and progesterone still remains low. In the luteal phase progesterone nears its maximum (Reimers, Büchel & Diekhof, 2014), whereas in the early follicular phase E2 and progesterone are at their nadir (Diekhof, 2015). Reimers, Büchel & Diekhof (2014) concluded that heightened E2 during the late follicular phase impaired the ability to slow down for reward maximization (NoGo learning ability), as opposed to the ability to speed up for higher reward (Go learning capacity). Diekhof (2015) extended these findings by showing a positive correlation between E2 and the ability to speed up for reward during the early follicular phase. This latter study indicated a better Go vs. NoGo learning ability during the early follicular phase and assumed that the boosting influence of the still increasing, yet intermediate E2 on dopamine probably optimally promotes Go learning ability.

Regarding the impact of testosterone on reward processing and reinforcement learning, data from humans are currently sparse. Also, rodent studies provide inconsistent findings about the influence of testosterone on reward processing. It has been observed that testosterone administration enhanced tyrosine hydroxylase (the rate-limiting enzyme catalyzing dopamine synthesis) in the substantia nigra of gonadectomized adolescent male rats (Purves-Tyson et al., 2012). Yet, testosterone may reduce tyrosine hydroxylase in gonadally intact adolescent male rats in the caudate putamen (Wood et al., 2013). Further, testosterone administration in gonadectomized adolescent male rats enhances mRNA of the dopamine degrading enzymes catechol-O-methyltransferase and monoamine oxidase in the substantia nigra (Purves-Tyson et al., 2012). In contrast, testosterone led to a significant increase of dopamine in the nucleus accumbens and dorsal striatum of gonadally intact male rats. Finally, in humans testosterone has been found to enhance striatal activity in the context of reward processing, while it decreased activation of the striatum during punishment processing (Morris et al., 2015).

Previous studies with early adolescents and young adults could not find a relation between testosterone and performance in cognitive or reward-related tasks (Halari et al., 2005; Ladouceur et al., 2019; White et al., 2020). Therefore, no clear assumptions can be made regarding the influence of testosterone on Go and NoGo learning. However, in light of its physiological significance for dopaminergic processing, a positive influence on reward processing and Go learning may be assumed.

Current study

In the present study, we assessed response time adjustments and learning behavior in the context of reward maximization in an adolescent sample. The salivary E2 and testosterone concentration was measured on the test day, which enabled us to examine the effect of the two sex hormones on Go and NoGo learning capacity. The adolescents performed an RT adjustment task, the so-called clock task (modified by Diekhof, 2015; created by Moustafa et al., 2008). In line with findings from adult research, we predicted that Go learning, associated with a better capability to speed up responding to maximize reward, would be related to “a higher E2 concentration” (e.g., Diekhof, 2015; Reimers, Büchel & Diekhof, 2014). Studies reporting a behavioral influence of testosterone on reward-related processing and especially reward learning are scarce. Whether higher testosterone would positively influence Go learning as well, could not be unconditionally hypothesized. Therefore, we examined the relation of testosterone and reinforcement learning capacity with the same analysis that was used to consider the impact of E2. Finally, we hypothesized that the effects of sex hormones on reinforcement learning would be different in female and male adolescents, mostly due to higher E2 concentrations in females and enhanced testosterone in males.

Materials & Methods

Participants

In total, 106 healthy German adolescents, between 11 and 18 years old, participated in this study. All participants had no history of psychiatric or neurological disorders and assured no regular medication intake. Fifteen adolescents were excluded from the analysis, because they showed a random response pattern throughout the task, which suggested that the task instructions had not been properly understood or that the respective participant lacked the motivation to perform the task properly. Another two participants were excluded because of technical problems that left the task unfinished. In sum, the data of 89 adolescents (mean age ± SD = 14.74 ± 1.9 years; 52 females) were analyzed.

Every participant had to sign a written declaration of informed consent before participation. In the case of minority, a legal guardian (parent) also had to sign a written declaration of informed consent before the testing. The adolescents were recruited in sports and other leisure clubs. The study protocol was approved by the local ethics committee of the Ärztekammer Hamburg (Ref: PV3948) and the study was conducted in accordance with “The Code of Ethics of the World Medical Association” (Declaration of Helsinki).

On the test day, participants were screened for depressive symptoms with the validated German Depression Inventory for Children and Adolescents (Stiensmeier-Pelster et al., 2014). Individual cognitive capacity was tested via the Digit-Span Test by measuring both forward and backward span from the German version of the Wechsler intelligence scale for childen (Wechsler, 2014) by counting the numbers that were correctly recalled. Self-reported trait impulsivity was examined with the German Version of the Barratt Impulsiveness Scale (BIS-11) for adolescents (Hartmann, Rief & Hilbert, 2011). Finally, every participant and the corresponding legal guardian filled out a translated version of the Pubertal Development Scale (PDS) (Petersen et al., 1988). We then calculated a mean of both scores and used it as an indicator of the degree of physical pubertal development of the given participant.

Experimental task

A modified version of the clock task (see Diekhof, 2015), that had been introduced by Moustafa et al. (2008) was used. In the task, three differently colored clock faces were presented. A full rotation of the clock arm lasted 5 s. Each clock face was assigned to one of three conditions, namely the fast, the random, and the slow condition. Each of the three clock conditions was shown 50 times in three sessions of 50 trials each, resulting in a total of 150 trials. The sequence of clock faces was pseudo-randomized and balanced for trial-type transitions (see also Diekhof, 2015 for further details on the clock task). The fast clock condition required a fast reaction once the clock arm started to move, in order to maximize reward outcome. The slow clock condition, in contrast, required the participant to postpone responding and slower RTs yielded higher reward. The random condition served as a control variant with no contingency between RT and reward outcome. It was used as an indicator of baseline response preference (see Fig. 1).

Figure 1 Task design.

(A) Reward was calculated using cosine functions for the fast and slow clock. A time-independent function for the random clock was applied as control condition. (B) Clock faces were presented pseudo-randomly for 5,000 ms. Once a button press was made, the clock arm stopped, and immediate feedback was given. After that, a blank screen was shown for the remaining time that the clock arm would have needed to complete the 5,000 ms. Therefore, the blank screen ensures a constant time duration of a trial. A trial ended with the achieved points presented for 1,000 ms.

The participants had to adapt to the optimal response speed in each condition to maximize their overall reward. The exact reward value of each trial in the fast and slow condition was calculated with a cosine function, ranging between a minimum of 15 and a maximum of 60 points. The random reward value was calculated with the difference between minimum and maximum points of reward multiplied by a random number and added with the minimum reward value (see Fig. 1). In every condition, a random noise parameter (range between −5 to +4 points) was applied to the reward. This was done to disguise the relation of a specific reward outcome with a specific RT. Immediately after the response, the reward outcome was shown to the participant. For the remaining time of a full clock arm turn, a blank screen was shown. Thus, each trial had the same length. If the participant did not respond within 5 s, no reward was presented, and the participant had to wait another 5 s before the next trial started.

Saliva collection and analyses

In the morning, three saliva samples were collected by the participant in 2 mL microcentrifuge tubes at home. Sample collection took place over the course of one hour (half-hourly samples) and started directly after awakening. The participants were allowed to drink water after the first sample up until 5 min before the second and third sample. They had to refrain from intake of food and beverages other than water during the sampling hour. Saliva samples were stored at −20 °C until further use. Before analysis, samples were thawed and centrifuged at room temperature at RCF 604 ×g (i.e., 3,000 rpm in a common Eppendorf MiniSpin centrifuge) for 5 min to separate the saliva from mucins. For the E2 analysis, a 17-β-Estradiol Saliva ELISA was used (Limit of Detection: 2.1 pg/mL), coated with anti-17-β-Estradiol antibody (monoclonal) with antibodies derived from donkey and sheep. For the testosterone analysis, a Testosterone Luminescence Immunoassay (both assays from Tecan/IBL International) was utilized (Limit of Detection: 1.8 pg/mL), coated with anti-mouse antibody. Intra-assay precision showed a mean CV of 8.8% (17-β-Estradiol Saliva ELISA) and 7.3% (Testosterone Luminescence Immunoassay). Inter-assay precision showed a mean CV of 11.8 (17-β-Estradiol Saliva ELISA) and 7.3% (Testosterone Luminescence Immunoassay).

The three morning samples were combined in an aliquot sample that consisted of an equal amount of saliva from every tube (100 μL). The analysis was done as described in the respective manual in our in-house laboratory. From the aliquot, two samples were assayed (n = 2). In addition, a high and a low control were analyzed. Subsequent behavioral analyses were done with standardized z-transformed values (zi=Xi−X ¯Sx) for each ELISA plate to standardize measurement inaccuracy of the plates.

Data preprocessing

For each subject, we calculated the mean RTs of each clock type. RTs under 200 ms were discarded, since they were very unlikely to reflect voluntary movements. In all, 125 trials (mean ± sd: 70 ± 72 ms) under 200 ms were excluded. We also calculated the mean RT of the initial 12 trials (called first block) and of the optimized last 12 trials (called last block) for each condition and participant (see Diekhof, 2015; Kohne et al., 2021; Moustafa et al., 2008; Reimers, Büchel & Diekhof, 2014 for a similar procedure). At the beginning of the experiment (in the first block), the participant did not know which clock face was associated with faster or slower responses for higher reward. Hence, the participant had to try to achieve the optimal outcome via various reactions exploring the task structure. Conversely, at the end of the clock task (in the last block), the participant should have been well adapted and was expected to show optimal RTs that led to the highest reward outcome in relation to individual clock faces.

Apart from the mean RT for the three clock types, the actual learning preferences that reflected individual Go and NoGo learning ability, respectively, were calculated from the last block. They reflected the adaption to the optimal response speed to the slow and fast clock, respectively, and allowed us to test the functional opponency of Go versus NoGo learning. For this, the RT of the slow and the fast clock were calculated in relation to the random clock, which provided information on the individual baseline response speed of a given participant. In order to calculate the optimized responses to the slow clock condition, we first subtracted the mean RT of the last 12 trials of the random clock condition from the mean slow clock RT of the last block. For standardization, this difference was then divided by the mean RT of the last 12 trials from the random clock. The resulting standardized relative RT reflects “optimized relative slowing”. Correspondingly, the subtraction of the mean fast clock RT from the mean random RT and its division by the mean random RT was used as the “optimized relative speeding” value.

The individual learning-related change in RT for each clock condition was calculated by subtracting the RT of the first block from the RT of the last block.

Data analyses

The behavioral data were analyzed with IBM SPSS Statistics 25. First, we performed a repeated measures General Linear Model (GLM) with the factors “clock condition” (fast, random, slow), “block” (first, last), “sex” (female, male) and “age” to test for possible effects of these factors on the RT. In another two GLMs the factor “age” was replaced by either the covariate “pubertal development” (PDS-score) or the z-standardized sex hormone concentration of E2 (zE2) and testosterone (zT) (see Results section below). This was done to assess the impact of pubertal maturation and sex hormones level on reinforcement learning. Post hoc tests used paired and independent t-tests, which were Bonferroni-corrected for multiple testing. If Levene’s test was significant, Welch’s t-test instead of Student’s t-test was used. The learning preference and effects of covariates were examined with a two-sided Pearson correlation. All effects and differences were considered as significant below a p-value of .05, two-tailed.

Results

Learning preference

Studies with adults revealed a reverse capability for adaptive speeding vs. adaptive slowing of responses in the clock task (Diekhof, 2015; Reimers, Büchel & Diekhof, 2014). Our data demonstrate that this reverse relation in adjustment preferences to either the slow or the fast clock may also exist in adolescents. We found that optimized relative speeding and slowing were negatively correlated in both sexes (females: r = −.48, p < .001; males: r = −.67, p < .001) (see Fig. 2). Adolescents who were better adjusted to the last block of the slow clock had difficulties to speed up for reward. In turn, participants who responded faster to the fast clock in the last block were impaired in the ability to slow down for reward.

Figure 2 Reverse relation of slowing and speeding.

Optimized relative speeding and slowing were negatively correlated in females, and males (p < .001).

General group characteristics

The female and male adolescents did not differ in their age, impulsivity (BIS-11), and zE2 concentration, which was determined by independent t-tests (see Table 1). The only significant differences between the two groups were a significantly higher zT level in males compared to females (t43.95 = −6.82, p < .001, d =  − 1.56) and a more advanced pubertal development of females compared to males (meanPDSfemales ± se: 3.03 ± .07; meanPDSmales ± se: 2.72 ± .09, t87 = 2.67, p = .009, d = .57).

Table 1 Group differences by sex.

	Females	Males	Females vs. males	
	Mean ± SD	n	Mean ± SD	n	t	p	95% CI	
							lower	upper	
Age (years)	14.67 ± 1.96	52	14.84 ± 1.83	37	−.4c	.689	−.98	−65	
zE 2	.14 ± 1.11	49	−.2 ± .56	35	1.59b	.177	−.09	.77	
E 2	5.89 ± 2.63 pg/mL	49	5.27 ± 2.08 pg/mL	35	.80	.425	−.64	1.49	
zT	−.53 ± .42	52	.74 ± 1.07	37	−6.82c	<.001	−1.64	−.89	
T	21.58 ± 14.1 pg/mL	52	89.61 ± 63.28 pg/mL	37	−6.43d	<.001	−89.45	−46.61	
BIS-11	63 ± 6.45	52	63.83 ± 9.57	36	−.46d	.65	−4.5	2.83	
PDS	3.03 ± .53	52	2.72 ± .56	37	2.67a	.009	.08	.55	
DICA	11.58 ± 6.37	52	9.39 ± 3.94	36	1.99e	.05	−.01	4.39	
Digit span forward	6.31 ± .9	52	6.31 ± .79	36	.01f	.991	−.37	.37	
Digit span backward	4.85 ± 1.29	52	4.89 ± 1.13	37	−.17a	.862	−.57	.47	
Notes.

a t87.

b t82.

c t43.95.

d t56.62.

e t85.09.

f t 81.25

g t38.55.

Influence of age and sex on response time adjustments

In an initial step, we assessed the influence of “chronological age” and “sex” of the participant on learning performance. For this, we used a repeated measures GLM including the covariate “age”, the between-subjects factor “sex” and the within-subject factors “clock condition” (fast, random, slow) and “block” (first, last). We solely found significant two-way interaction of “clock condition” x “block” (F2,172 = 4.41, p = .014, η2p = .05). This was reflected by a change in the RT from the initial to the optimized last block in the fast (t88 = 11.08, p < .001, d = 1.17, Bonferroni corrected for three comparisons) and in the slow condition (t88 = −13.79, p < .001, d =  − 1.46, Bonferroni corrected for three comparisons), but not in the random condition (t88 = .14, p = 1, d = .02, Bonferroni corrected for three comparisons) (Table 2).

Table 2 Comprehensive summary of RTs and post-hoc results.

		Mean RT ± SE	Females vs. males		Correlations of all participants	
Block	Clock	Females & males	Females	Males	t (df = 87)	p	95% CI	zT	zE 2	PDS	
							lower	upper	r	p	r	p	r	p	
first
& last	FAST	1264 ± 37msa,b,***	1302 ± 54 ms	1212 ± 293 ms	1.2	.234	−60 ms	241ms	−.12	.327	−.08	.497	−.12	.274	
	RANDOM	2196 ± 65 msa,c,***	2157 ± 562 ms	2253 ± 695 ms	−.71	.477	−360 ms	170ms	.23	.032 **	−.02	.843	.19	.068*	
	SLOW	3458 ± 67 msb,c,***	3346 ± 653 msd,**	3617 ± 593 msd,**	−2	.048 **	−539 ms	−2 ms	.28	.009 **	−.04	.731	.1	.359	
	ALL CLOCKS	2307 ± 333 ms	2269 ± 311 ms	2360 ± 360 ms	−1.28	.203	−234 ms	50 ms	.29	.007 **	−.09	.412	.14	.185	
first	FAST	1610 ± 58 msd,***	1655 ± 571 ms	1547 ± 524 ms	.92	.363	−127 ms	345 ms	−.08	.441	−.24	.03 **	−.12	.249	
	RANDOM	2203 ± 78 ms	2104 ± 685 ms	2343 ± 794 ms	−1.52	.132	−552 ms	73 ms	.18	.084*	.08	.469	.1	.354	
	SLOW	2945 ± 87 mse,***	2791 ± 807 ms**	3163 ± 803 ms**	−2.15	.034 **	−717 ms	−29 ms	.3	.004 **	−.06	.572	.1	.366	
last	FAST	919 ± 36 msd***	949 ± 382 ms	877 ± 275 ms	1.04	.3	−74 ms	219 ms	−.04	.718	.1	.383	−.04	.692	
	RANDOM	2190 ± 80 ms	2211 ± 703 ms	2162 ± 834 ms	0.3	.765	−276 ms	374 ms	−.14	.195	−.12	.275	.22	.038 **	
	SLOW	3972 ± 66 mse***	3902 ± 694 ms	4071 ± 503 ms	−1.33	.188	−435 ms	97 ms	.23	.03 **	<.01	.991	.07	.491	
Notes.

Equal letters mean significant paired t-Test results.

*** p < .001.

** p < .05.

* p < .1.

a t88 = −12.51; 95 CI −1080 ms, −784 ms.

b t88 = −25.93; 95 CI −2362 ms, 2026 ms.

c t88 = 15.2; 95 CI 1097 ms, 1427 ms.

d t88 = −11.08; 95CI −815 ms, −567 ms.

e t88 = 13.79, 95 CI 879 ms, 1175 ms

Influence of pubertal development and sex on response time adjustments

The first GLM was repeated with the factor “pubertal development” (measured with the PDS) replacing the factor “age”. A significant main effect of  “clock condition” (F2,172 = 7.28 p = .001, η2p = .08), significant two-way interactions of “clock condition” x “pubertal development” (F2 = 3.4, p = .036, η2p = .04) and “clock condition” x “sex” (F2 = 3.81, p = .024, η2p = .04) emerged. Further, the interaction between “clock condition” and “block” remained significant (F2,172 = 8.04, p < .001, η2p = .09).

Post hoc t-tests showed a significant RT distinction between the three clock conditions (fast vs. random: p < .001, d = −1.33; fast vs. slow: p < .001, d =  − 2.75; slow vs. random: p < .001, d = 1.61, Bonferroni corrected for two comparisons) (see Table 2). Consequently, an adjustment to the varying clock conditions could be observed. Concerning the interaction between “clock condition” and “sex”, a significant difference only arose in the slow clock condition. Males reacted significantly slower and thereby better to the slow clock in general than females did (p = .048, d = −.43) (see Table 2). The interaction of “pubertal development” and “clock condition” was reflected by a trend-wise positive correlation between the PDS and the RT of the random condition only (r = .19, p = .068) (see Table 2).

Influence of sex hormones and sex on response time adjustments

In a third GLM we investigated the modulatory influence of zE2 and zT as a function of the participants’ sex on RTs in the three clock conditions (fast, random, slow) and the two blocks (first, last). The main effect of “clock condition” (F2,160 = 114.83 p < .001, η2p = .81) and the interaction of “clock condition” and “block” (F2, 160 = 7.28 p < .001, η2p = .59) remained significant. Furthermore, an interaction of “block x clock condition x zE2 concentration” (F2 = 4.9, p = .009, η2p = .06) and a main effect of block (F1,80 = 5.29 p = .024, η2p = .06) and of “zT” (F 1 = 5.28 p = .024, η2p = .06) occurred.

The interaction of “block x clock condition x zE2” was reflected by a negative correlation between zE2 and the initial RT in the fast clock condition (r = -.24, p = .03) (see Fig. 3). In addition, we also examined the individual learning-related change in the RTs between first and last block, which demonstrated the adjustment from the initial to the optimized block (RT last block –RT first block). The learning-related change showed a significant positive correlation with zE2 in the fast clock condition (r = .28, p = .01) (see Fig. 4). No correlation emerged with the slow (r = .08, p = .497) or random condition (r = −.18, p = .096).

Figure 3 Negative correlation between zE2 and the initial fast clock.

Subjects who had higher zE2 concentrations responded faster during the initial fast clock condition (r = −.24, p =.03).

Figure 4 Positive correlation between zE2 and the learning-related change of the fast clock.

Subjects who had lower zE2 concentrations showed a better adjustment from the initial to the optimized block in the fast clock condition, and became relatively faster in the last block, which resulted in as indicated by a more negative delta value of “last - first block” (r = .28, p = .01).

A post-hoc comparison of the blocks evinced a slower response speed in the initial block compared to the last block (t 88 = −2.67, p = .009, d = −.28). Further, zT was positively correlated with a slower RT independent of clock condition or block (r = .29, p = .007) (see Fig. 5). Since we found a significant difference in the zT of females and males, with higher concentrations in males (see Table 1), we additionally explored the zT effect separately for both sexes. From this, it became obvious that the correlation probably emerged from the male adolescents. Accordingly, the mean of both blocks across all clocks was positively correlated with zT in males (r = .48, p = .002), but not in females (r = −.15, p = .298). In males, a general slowing could also be observed with increasing zT in both blocks of all conditions (first: r = .37, p = .025, last: r = .5, p = .002) and especially in the slow (r = .42, p = .01) and the random (r = .35, p = .032), but not in the fast condition (r = .09, p = .579). Additionally, in the initial (r = .35, p = .036) and optimized block (r = .44, p = .007) of the slow clock positive correlations emerged. Again, these correlations could not be found in females.

Figure 5 Positive correlation between zT and the response time of all clocks and both blocks.

Subjects who had higher zT concentrations generally responded more slowly (r = .29, p = .007).

Discussion

This study examined the effects of adolescent E2 and testosterone concentrations on RT adjustments in the clock task. Results indicate individual differences in the preference for either Go or NoGo learning (see Fig. 2) and an adaption to the different clock conditions from the initial to the optimized block. Both findings have already been demonstrated previously in studies with adults (Kohne et al., 2021; Moustafa et al., 2008; Reimers, Büchel & Diekhof, 2014). In addition, we also found that testosterone levels were significantly higher in males then females, while age, impulsivity and E2 concentrations did not differ between the sexes. We also did not observe an age-dependent influence on the RT, and there was no association between individual pubertal development and Go or NoGo learning. Solely, a tendency towards a slower baseline response speed with increasing pubertal development emerged. Apart from that, we found a sex difference in the slow clock condition. Male adolescents responded significantly slower (better adapted) to the slow clock condition compared to females. E2 and testosterone further appeared to modulate learning ability in different ways. Whereas E2 apparently enhanced initial Go learning (see Figs. 3 and 4), testosterone presumably promoted NoGo learning ability (see Fig. 5), yet primarily in males.

Similar to studies with adults, our data confirmed the detection of a preference for Go or NoGo learning ability with a presumable supporting effect of E2 on Go learning (Diekhof, 2018; Moustafa et al., 2008; Reimers, Büchel & Diekhof, 2014). Furthermore, we observed a relation between habitual testosterone and the ability to slow down for reward, which was especially evident in male adolescents. The observed divergence of females and males in the learning capability related to the slow condition could probably be ascribed to a hormonal sex-difference. Hormonal testosterone concentrations differed significantly between females and males who showed enhanced concentrations. The varying increase of gonadal hormones during puberty could thus be one reason for the different RT adjustments in the slow clock. Accordingly, testosterone was associated with a slower RT and enhanced NoGo learning in adolescents. An explorative analysis showed that this result could be traced back to the male adolescents, most likely because testosterone is the main acting gonadal hormone during male pubertal development and by far more variable in pubertal males than in females. In line with adult research, E2 seemed to stimulate the initially faster responses and therefore Go learning in all adolescents. We speculate that the effect of E2 could have been mediated by its modulatory impact on dopaminergic transmission, which has been assumed for similar findings in adult women (see i.a. Diekhof, 2015; Reimers, Büchel & Diekhof, 2014). Estrogen receptors can be found in the brain of both sexes via which E2 presumably has modulating effects on neurotransmission and plasticity (Gillies & McArthur, 2010).

The correlation between Go learning and E2 occurred exclusively in the initial block during which participants were still naïve regarding the temporal reward associations of the different clocks. This might indicate that E2 has only a subtle effect on behavioral responding in the clock task. Once the RT had been optimized in later phases of the task, this correlation was no longer behaviorally measurable (see also Reimers, Büchel & Diekhof, 2014).

Alternatively, E2 may also support learning through a promotion of signal transduction. E2 administration in young and aged ovariectomized rhesus monkeys led to an increase in spine density in the dorsolateral prefrontal cortex (Hao et al., 2003). An increased spine density on pyramidal neurons is connected to an enhanced number of excitatory synapses per neuron which in turn might improve learning performance in general (Mahmmoud et al., 2015). Moreover, in ovariectomized rats E2 administration provoked cell proliferation and an increase of dendritic spine density in the hippocampus (Adams et al., 2002; Tanapat, NB & Gould, 2005). In a previous study, Davidow and colleagues demonstrated the positive impact of hippocampal activity and its connectivity to the striatum on reinforcement learning in adolescents (Davidow et al., 2016). Therefore, the potentiating influence of E2 on the hippocampus may improve reward learning as well. Besides E2, androgens also positively affect prefrontal and hippocampal processing, but rat studies indicate a greater impact of androgens in males (Hamson, Roes & Galea, 2016).

Similar to E2, testosterone can modulate dopaminergic transmission and may also impact transmission in other neurotransmitter systems (De Souza Silva et al., 2009; Sinclair et al., 2014). The enhancing effect of testosterone on slowing ability may additionally be explained through an interaction of testosterone and serotoninergic processing in males. In male rats, testosterone administration leads to an increase of cerebral serotonin and its metabolites (De Souza Silva et al., 2009; Thiblin et al., 1999). Moreover, a positive correlation between plasma testosterone and serotonin receptor 4 level emerged, leading to the suggestion that higher testosterone is accompanied by a higher cerebral serotonin tonus (Perfalk et al., 2017). Therapeutic approaches include selective serotonin reuptake inhibitors that increase synaptic serotonin levels and modulate neuroplasticity (Kraus et al., 2017). For learning and memory formation synaptic plasticity is exceedingly important. Serotoninergic impact on human behavior and neurophysiological processes is commonly investigated through a depletion of the serotonin precursor tryptophan. Studies with healthy humans using tryptophan depletion demonstrate a slowing of responses by pharmacologically increased serotonin (e.g., (Murphy et al., 2002)). We observed a better slowing ability with habitually increased testosterone, which might indicate that this could have been an indirect effect of testosterone on serotoninergic transmission. This would also be in line with other studies, that found that the effect of behavioral slowing in punishment contexts, especially under high incentive motivation, disappeared, if serotonin was pharmacologically depressed (e.g., Crockett et al., 2012). Lowered serotonin concentrations after depletion have further been associated with decreased neural sensitivity to punishment (Helmbold, Zvyagintsev & Dahmen, 2015). Hence, enhanced testosterone concentration might have driven NoGo learning and enabled a better slowing down for reward, through its interaction with the serotoninergic system.

Just as a recent study, we could not observe a relation between reward or punishment sensitivity and the pubertal stage (Chahal et al., 2021). A generally lowered response speed in further developed adolescents could be a consequence of reduced impulsivity, which may be an indicator of neurophysiological and cognitive maturation. Similar to others, we did not find an association with chronological age (Wierenga et al., 2018). Our results thus support the assumption that pubertal development is a better indicator regarding cognitive performance than chronological age.

To date, a non-invasive direct measurement of neurotransmitter processes like dopamine binding or synthesis in the adolescent human brain is not feasible. We used non-invasive measurements to determine steroid hormone concentrations and assessed the individual learning ability for Go and NoGo learning. By combining both parameters, we tried to apply them as indirect indicators of dopaminergic transmission. Besides E2 and testosterone other steroid hormones are presumably attractive for future studies. For instance, the influence of progesterone as a counterpart to E2 on dopaminergic action may be of increased future interest. Whereas E2 is assumed to have an agonistic effect on dopaminergic transmission, progesterone supposedly reduces E2 receptor density (Selcer & Leavitt, 1988) and apparently upregulates monoamine oxidase when it is administered together with E2, which mimics the luteal phase of a natural menstrual cycle (Luine & Hearns, 1990; Luine & Rhodes, 1983). Additionally, progesterone enhances gamma-aminobutyric acid induced inhibition of dopaminergic neurons (Majewska et al., 1986). Thus, an antagonistic and reducing effect of progesterone on dopaminergic transmission has been suggested (Diekhof, 2018). In future studies, the tracking of the developing menstrual cycle of the female adolescents could probably contribute to a better interpretation of the opposite effects of E2 and progesterone.

Finally, genetic predisposition as such has already been observed to affect reward sensitivity (Richards et al., 2016), and may further interact with steroid hormone level as demonstrated previously (Jakob et al., 2018; Veselic et al., 2021). In addition to previous findings on receptor and transporter polymorphisms of dopamine, serotonin and sex hormones, future studies could examine genetic interactions via genome-wide associations.

Conclusion

Sex hormones modulate neurophysiological processes and behavior in the context of reward processing in both adult animals and humans. However, evidence from adolescent populations is sparse. The present study assessed the impact of E2 and testosterone on adolescents’ reinforcement learning. Similar to female adults (e.g., Diekhof, 2015), E2 promoted initial Go learning in both sexes in our adolescent sample. Testosterone, in turn, enhanced NoGo learning in males. It could be speculated that individual differences in reinforcement learning are associated with variations in these hormones during adolescence, which shift the balance between a reward and avoidance-related learning style.

Future investigations should consider further steroid hormones (e.g., cortisol, progesterone) and neurophysiological processing to specify the impact of hormonal differences on the dopaminergic mechanisms of reinforcement learning.

Supplemental Information

Supplemental Information 1 Prepared data

Click here for additional data file.

Supplemental Information 2 Raw data

Click here for additional data file.

The authors would like to thank Angelika Kroll for her support in laboratory analyses.

Additional Information and Declarations

Competing Interests

Author Contributions

Human Ethics

Data Availability

The authors declare there are no competing interests.

Sina Kohne conceived and designed the experiments, performed the experiments, analyzed the data, prepared figures and/or tables, authored or reviewed drafts of the paper, and approved the final draft.

Esther K. Diekhof conceived and designed the experiments, analyzed the data, authored or reviewed drafts of the paper, and approved the final draft.

The following information was supplied relating to ethical approvals (i.e., approving body and any reference numbers):

This study was approved by the local ethics committee of the Ärztekammer Hamburg (Ref: PV3948).

The following information was supplied regarding data availability:

The raw measurements are available in the Supplementary File.

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
