# Peer review of "Testosterone and estradiol affect adolescent reinforcement learning"

_PeerJ, doi:10.7717/peerj.12653_

## Round 0.1 · original submission · Major Revisions

Reviewers have commented on your work, and have raised very serious concerns. One recommended reject, the other did not. The editor would like to give authors the opportunity to address all concerns raised. Thank you.

·

Basic reporting

Thank you for considering me as reviewer for PeerJ.

Professional English language:
The authors’ language is clear and understandable (although it sounds rather like German). However, the English language is not as professional as I would expect it to be in a scientific manuscript. Tenses are not always used correctly (e.g. future tense in line 56 f.; line 389: “we could confirm”-> we confirmed), sentences are rather long, and, in general, the language sounds colloquial (e.g. line 242). I suggest you have a colleague who is proficient in English and familiar with the subject matter review your manuscript. (I am no native speaker either)

Intro & Background:
After I have read the title and abstract, I expect to read a manuscript about hormonal influences on adolescent reinforcement learning behavior. The Introduction however, describes the neurobiological background of reward learning very profoundly. I understand that you need to describe the background of the dopaminergic reward system a little bit, but I think you go into this too deeply. I recommend to focus more on studies describing hormonal effects on reward processing in general, with emphasis on behavioral work in humans. I know that there is not much literature about this topic, but here you could stress why your work is a real contribution to the field.
And also, if you did not measure DA and 5-HT, do not put too much emphasis on it in the first place. I would be careful with simply stating that your task can be used as a proxy for DA and 5-HT.

Structure conforms to PeerJ standards and discipline norm. -> yes

Figures are relevant, well labelled and described. The figure legends in Figure 2 and 3 are very small.

Raw data supplied -> yes

Experimental design

Original primary research within scope of the journal -> yes

The research question is interesting and meaningful for understanding the development of certain behaviors and, on a more general level, typical mental disorders in adolescents (depression, substance use disorder, ADHD etc.). However, that’s what I would like to read a little bit more in Introduction and Discussion. The interesting part of your study is how you’ve tried to understand a possible effect of sex hormones on reward learning during pubertal age – because you can assume that hormones critically affect behavior during puberty.

Rigorous investigation performed to a high technical & ethical standard. -> yes

Methods described with sufficient detail and information to replicate. -> yes

Validity of the findings

All underlying data have been provided; they are robust, statistically sound, and controlled. -> yes. Data analysis and results are reported sufficiently. The analysis methods are chosen correctly to answer the research questions. Effect sizes are reported.

Discussion and Conclusions:
See my comments on the Introduction. I advise against describing to many neurobiological details. Especially lines 346-358 and lines 406-419 could be densed and rewritten for readers with less neurobiological background, i.e. a broader audience.

Additional comments

- Did you check for the onset of menses in girls? If they already had their menses, in which menstrual cycle phase did you measure? This could have a major impact on interpreting your results (see also Diekhof 2015).
- What do you mean with “optimal hormonal concentrations” (line 152)? Is there an optimal hormonal level? Please describe.
- I recommend not using too many abbreviations. It hinders the reading flow. At least, I would fully spell out testosterone and estradiol. But that is a rather personal opinion.

Reviewer 2 ·

Basic reporting

Basic reporting:
1. English language should be improved throughout to ensure that ideas are clearly communicated. Current examples of awkward phrasing include line 30, 39-40, 42-44, 123-125, 126-129, 133-135 and several others. Current phrasing makes comprehension difficult.
2. Line 32- Units of age should be specified (years).
3. Introduction
a. Primary issue:
i. The structure of the introduction is not intuitive. Why does it start with such a specific focus on dopamine when neurotransmitter levels are not even measured in this study? It would seem to make more sense to focus on the context of the results presented. For example, it sounds like the take away message is that sex hormones have common and dissociable sex-dependent effects on different aspects of reinforcement learning. Therefore, it would seem that an introduction covering sex differences in cognition during adolescence and the moderating effects of estradiol and testosterone would be more helpful in orienting the reader. Such an extensive discussion of dopamine and serotonin would be more appropriate for the discussion.
b. Other comments on the introduction:
i. In the paragraph on steroid hormone interactions with dopamine and serotonin, there is no mention of the extensive literature from human subjects that provide additional translational support for such interactions.
ii. The statement at line 112 is inaccurate. The authors state, “In adolescents, a rising hormonal level contributes to pubertal development, whereas in adults a constant sex hormone level maintains reproductive function.” However, adult women do not have constant hormone levels. Also, maintaining constant estrogen/progesterone levels would suppress ovulation.
iii. The citation at line 126 should be “Jacobs & D’Esposito, 2011” not “Cools & D’Esposito, 2011”
4. The structure of the discussion is similarly confusing and circuitous.

Experimental design

Experimental design:
1. Was menstrual cycle phase accounted for? This also would affect DA and 5-HT
2. Why were RTs under 200 ms discarded? How many data points was this? Was this throughout blocks or just the first trial?
3. Age should be included as a covariate in the primary statistical model evaluating sex. Age effects should also be controlled for when examining pubertal effects, either by including it as a covariate if not co-linear or by first regressing age effects and performing further analyses on residuals. Although there is no significant difference in age between male and female groups (though arguably, age should still be controlled for in this case given that this is a developmental sample), the authors also examine other factors such as hormones and pubertal effects. Not controlling for age in a developmental sample is a fundamental flaw.
4. The authors interchangeably use terms for sex and gender. Terms should be consistent. If sex was studied, terms should be sex, male, and female. If gender was studied, terms should be gender, boys, and girls.
5. Given the current use of language, it is unclear if the methods are described in sufficient detail to replicate.

Validity of the findings

Validity of Findings
1. Data have not been provided in a repository.
2. The paragraph under “conclusion” seems only tangentially related to actual results. It would be helpful to have a sentence that clearly links results to conclusions to ground readers.

Additional comments

In this manuscript, Kohne and Diekof examine correlations between endogenous hormones and reinforcement learning in male and female adolescents. They found associations between markers of reinforcement learning and levels of estradiol and testosterone. While this is an important topic, and literature regarding hormone and pubertal effects on cognitive development are essential to furthering our understanding of normal and abnormal brain development, my enthusiasm is diminished by the structure of the manuscript and awkward use of English.

---

## Round 0.2 · Minor Revisions

Please, authors, your manuscript is very promising. Do kindly attend to the comments raised by the reviewer. Look forward to your revised manuscript.

·

Basic reporting

The English language has much improved and sounds professional now.

Intro & Background:
My comments regarding Introduction and Background have been adressed sufficiently. The reader can follow the thoughts of the authors and is able to understand the rationale of the study.

Structure conforms to PeerJ standards and discipline norm. -> yes

The figures are very small in the PDF file. I hope this is just a copy and paste issue and will be restored during production?

Raw data supplied -> yes

Experimental design

Original primary research within scope of the journal -> yes

Research question well defined, relevant and meaningful. It is stated how the research fills an identified knowledge gap. -> yes, all my comments have been adressed.

Rigorous investigation performed to a high technical & ethical standard. -> yes

Methods described with sufficient detail and information to replicate. -> yes

Validity of the findings

all my comments have been adressed.

Additional comments

all my comments have been adressed.

·

Basic reporting

The authors present a valuable study on understanding the effect of testosterone and estradiol on adolescent reinforcement learning. Data are well presented, however, authors need to amend a few parts of the English language. Confusion in understanding the context can be observed in certain part of paragraphs.

Experimental design

Please refer to the main article.

Validity of the findings

Please refer to the main article.

---

## Round 0.3 · accepted · Accept

Thank you authors for revising your work. It is now acceptable for publication. Thank you for finding PeerJ as your journal of choice. Look forward to your future scholarly contributions to PeerJ.
Congratulations and very best regards.